# Novel carbon skeletons activate human NicotinAMide Phosphoribosyl Transferase (NAMPT) enzyme in biochemical assay

Karen H. Almeida[ID]*, Lisbeth Avalos-Irving[¤a◕], Steven Berardinelli[¤b◕], Kristen Chauvin[¤c◕], Silvia Yanez[¤a]

Physical Sciences Department, Rhode Island College, Providence, Rhode Island, United States of America

◕ These authors contributed equally to this work.
¤a Current address: Quality Control Department, Amgen Inc., West Greenwich, Rhode Island, United States of America
¤b Current address: Department of Biochemistry and Molecular Biology, University of Georgia, Athens, Georgia, United States of America
¤c Current address: Department of Biology and Institute of Neurovscience, University of Oregon, Eugene, Oregon, United States of America
* kalmeida@ric.edu

**Data Availability Statement:** All relevant data are within the paper and its Supporting Information files.

## Abstract

Nicotinamide adenine dinucleotide (NAD) is a central molecule in cellular metabolism that has been implicated in human health, the aging process, and an array of human diseases. NAD is well known as an electron storage molecule, cycling between NAD and the reduced NADH. In addition, NAD is cleaved into nicotinamide and Adenine diphosphate ribose by NAD-consuming enzymes such as sirtuins, PARPs and CD38. There are numerous pathways for the biosynthesis of NAD to maintain a baseline concentration and thus avoid cellular death. The NAD salvage pathway, a two-step process to regenerate NAD after cleavage, is the predominant pathway for humans. Nicotinamide PhosphribosylTransferase (NAMPT) is the rate-limiting enzyme within the salvage path. Exposure to pharmacological modulators of NAMPT has been reported to either deplete or increase NAD levels. This study used a curated set of virtual compounds coupled with biochemical assays to identify novel activators of NAMPT. Autodock Vina generated a ranking of the National Cancer Institute's Diversity Set III molecular library. The library contains a set of organic molecules with diverse functional groups and carbon skeletons that can be used to identify lead compounds. The target NAMPT surface encompassed a novel binding location that included the NAMPT dimerization plane, the openings to the two active site channels, and a portion of the known binding location for NAMPT substrate and product. Ranked molecules were evaluated in a biochemical assay using purified recombinant NAMPT enzyme. Two novel carbon skeletons were confirmed to stimulate NAMPT activity. Compound 20 (NSC9037) is a polyphenolic xanthene derivative in the fluorescein family, while compound 2 (NSC19803) is the polyphenolic myricitrin nature product. Micromolar quantities of compound 20 or compound 2 can double NAMPT's product formation. In addition, natural products that contain high concentrations of polyphenolic flavonoids, similar to myricitrin, also stimulate NAMPT

**Funding:** Research reported in this publication was supported by grant subawards to KHA from the Rhode Island Institutional Development Award (IDeA) Network of Biomedical Research Excellence (https://web.uri.edu/riinbre/) from the National Institute of General Medical Sciences of the National Institutes of Health (https://www.nigms.nih.gov/) under grant number P20GM103430 and from the National Center for Research Resources of the National Institutes of Health under grant number 5P20RR016457-11. Its contents are solely the responsibility of the authors and do not necessarily represent the official views of the NIGMS or the NIH. Furthermore, this research is based in part upon work conducted using the Rhode Island Genomics and Sequencing Center that is supported in part by the National Science Foundation (https://www.nsf.gov/) under EPSCoR Grants Nos. 0554548 & EPS-1004057. The funders had no role in study design, data collection and analysis, decision to publish, or preparation of the manuscript. There was no additional external funding received for this study.

**Competing interests:** The authors have declared that no competing interests exist.

activity. Confirmation of a novel binding site for these compounds will further our understanding of the cellular mechanism leading to NAD homeostasis and better human health outcomes.

## Introduction

Nicotinamide Adenine Dinucleotide (NAD) is a critically important energy molecule within living systems. NAD performs numerous roles within the cell, the most widely recognized is its role as a universal electron carrier. NAD is a coenzyme, electron-storage molecule for hundreds of oxidoreductase enzymes that either remove electrons from or deliver electrons to substrates, thus allowing NAD to cycle between its oxidized and reduced (NADH) structures. The most widely studied example of this cycling is cellular respiration when during glycolysis and citric acid cycle, electrons are removed from the glucose carbon skeleton and stored as NADH. NADH then delivers those electrons into the electron transport chain to drive the generation of the proton gradient which in turn powers the formation of ATP from ADP and Pi by ATP synthase [1].

NAD is also used as a substrate by NAD-consuming enzymes, such as PARPs, sirtuins and CD38. The discovery of NAD-consuming enzymes implicated NAD in a broader set of cellular functions, including cell signaling, DNA repair, cell division, epigenetics, and the normal aging process. For NAD-consuming enzymes, the structure of NAD is cleaved into two product molecules, nicotinamide (NAM) and ADP-ribose. Cellular conditions warranting the hyperactivity of these enzymes depletes NAD levels, which is implicated in multiple disease states [2]. An example is Poly-(ADP ribose) polymerase 1, PARP1, which uses NAD to ADP-ribosylate itself (among other targets) as a recruitment tool for the repair of damaged DNA. Multiple NAD molecules are required to initiate this repair, thus significant damage requires overactivation of PARP1 that then can deplete NAD cellular concentrations and compromise basic cellular functions, eventually leading to cell death [3]. The requirement of NAD in fundamental cellular functions suggests that a baseline concentration of NAD is essential and therefore NAD homeostasis is tightly regulated.

Given the significance of NAD, there are multiple pathways to maintain sufficient concentrations of NAD. NAD can be generated from tryptophan, nicotinic acid, or NAM. The predominant pathway within mammals is the NAD salvage pathway [4]. This is a two-step process that starts with NAM, the product of NAD-consuming enzymes, Fig 1. The enzyme nicotinamide phosphoribosyltransferase (NAMPT) condenses NAM with phosphoribosyl pyrophosphate (PRPP) to generate nicotinamide mononucleotide (NMN). This condensation requires the hydrolysis of an ATP molecule to phosphorylate H247 of NAMPT [5]. NMN is then converted to NAD by the enzyme family of Nicotinamide mononucleotide adenylyltransferase 1–3 (NMNAT1-3). The NAD salvage pathway scavenges NAM from the cell, converting it to the more ubiquitous NAD molecule. NAMPT, the enzyme that controls the overall rate of the pathway, is a homodimeric type II phosphoribosyltransferase in which dimerization generates two active site channels [6].

NAD and precursors within NAD biosynthetic pathways are implicated in many human illnesses as well as the normal aging process [2]. Molecular modulation of the activity of NAD synthesizing enzymes is also under investigation [7, 8]. The first highly specific and potent inhibitor of NAMPT, designated FK866 or APO866, was identified in 2003 [9]. FK866 has structural overlap with NAMPT's natural substrate nicotinamide NAM and is known to bind within the active site channels to progressively exhaust cells of NAD, resulting in cell death

**Fig 1. Schematic of NAD+ biosynthetic pathway and NAMPT enzymatic assay.**

[10]. Since the identification of the FK866 binding site, there has been many structures reported to inhibit NAMPT. However, clinical trials on NAMPT inhibition suggest that the toxicity of these agents when used individually may outweigh their benefit. Studies to determine the efficacy of NAMPT inhibition in combination with established drug regimens, as a means of augmenting their effect, are under investigation [11]. Reviews of NAMPT inhibition suggest a diverse set of carbon structures are able to limit NAMPT biochemical activity. Nevertheless, the known or theorized binding locations significantly overlap with the active site channels [12, 13]. Radically diverse carbon structures have also been reported to inhibit NAMPT, although the $IC_{50}$ values are significantly higher than FK866 indicating far less inhibition and again these molecules are thought to bind within the active site channels [14, 15].

The first reported activator of NAMPT was P7C3 [16]. Within the last two years, several reports have been published that identify small molecule activators of NAMPT. SBI-797812 was isolated from a HTS of over 57,000 compounds and induced a concentration-dependent stimulation of NMN production in the presence of NAM, PRPP and ATP [17]. Interestingly, SBI-797812 is structurally similar to known NAMPT inhibitors that bind in the active site channels. Site-directed mutants of NAMPT (G217R) that decrease FK866 inhibitor binding also decrease the stimulator effect of SBI-797812 family members, suggesting an overlapping binding site within the active site channel [17, 18]. The interaction of SBI-797812 to NAMPT is thought to dramatically shift the equilibrium in favor of NAM consumption, ie NMN product formation, by shielding the essential phosphorylated histidine 247 residue within the active enzyme. However, Yao *et al* report that SBI-797812 inhibits NAMPT at concentrations below 10uM NAM but stimulates NAMPT activity at higher concentrations. They attribute the activation to the relieving of substrate inhibition conditions and identify another NAMPT activating molecule for consideration [19]. A HTS biochemical screen for NAMPT activators

revealed the lead compound 2-(2-*tert*-butylphenoxy)-*N*-(4-hydroxyphenyl)acetamide, called NAT for NAMPT activator. Compound optimization yielded a family of activators and X-ray crystallography of NAT bound to the NAMPT dimer confirms the binding site for NAT family members within the active site channel and overlapping with the active site residues. NAT protects U2OS cells from FK866 mediated cell death, identifying K189 as a residue critical for strong binding of the activators [19]. However, structure-activity studies between NAMPT and a family of small molecule modulators suggests that increased product formation may be achieved through a critical water mediated interaction within the NAMPT active site but independent of K189 [20]. Clearly the mechanism of NAMPT activation is still an open investigation. Finally, both P7C3 derivatives and NATs provide a neuroprotective effect in chemotherapy-induced peripheral neuropathy mouse models, likely through an enhanced NAD production [19, 21].

Herein we report the virtual screening of the NCI Diversity set III library that identified two novel carbon skeletons able to activate NAMPT in biochemical studies. Computational screening suggests a unique binding location that does not overlap with NAMPT active site channels and thus may provide an alternative mechanism for activation.

## Materials and methods

### Virtual screening

Steps involved in the virtual screening of potential NAMPT inhibitors are briefly described below.

**Preparation of ligand library.** The National Cancer Institute's Open Chemical Repository Diversity Set III was downloaded from the NCI website (http://dtp.nci.nih.gov/branches/dscb/div2_explanation.html) in SMILES format (.smi). Ligands containing salts or Arsenic atoms were removed from the library prior to docking with the target. Library files were further processed to add hydrogens and converted from 2D (.smi) files to 3D (.sdf) files using the freeware Marvin Beans. FK866 was incorporated into the Diversity Set III library as a known binding control within the active site. At the time of these experiments, there were no known activators of the NAMPT enzyme.

**Preparation of enzyme target.** The coordinates of human NAMPT complexed with the reaction product nicotinamide mononucleotide (NMN; PDB ID: 2GVG) or the inhibitor n-(4-(1-benzoylpiperidin-4-yl)butyl)-3-pyridin-3-ylpropanamide (FK866; PDB ID: 2GVJ) were obtained from the Protein Data Bank (PDB). (http://www.rcsb.org/pdb/home/home.do) The protein file was prepared for docking by removal of water molecules, addition of polar hydrogens using the freeware MolProbity and removal of ligands from the active site. A grid box of 25 cubic angstroms that encompassed the NMN and FK866 binding sites was constructed and used to verify docking accuracy. NMN and FK866 were independently redocked into the prepared target NAMPT and assessed for accuracy against the original crystallized structures [10]. RMSD values were calculated in excel and determined to be below the 2 Å threshold, indicating similar positioning.

**Library docking.** The target search space (25 cubic Å) for docking the library was constructed around the K389, S199, and K229 residues. Docking was completed through the PyRx freeware, a Graphic User Interface for Autodock Vina. Autodock Vina outputs the nine best docking configurations for each ligand within the target search space. Configurations with the highest binding affinity were ranked and sorted based on their overall affinity to the target region. A random search space was constructed and docked with the identical library to extract compounds with a high likelihood of non-specific interactions. Binding energies of the known inhibitor, FK866, within the active site was used as a cutoff binding affinity, generating a large

pool of potential modulators to screen. All molecules binding at the determined cutoff binding energy and appearing on all search lists were eliminated as potential lead compounds.

## Reagents and plasmid generation

All reagents were purchased through Sigma Aldrich unless specified otherwise. Gateway entry plasmid for human NAMPT cDNA (pENTR) was a kind gift from Dr. Robert W. Sobol, (Brown University, Providence, RI). Sequence was confirmed on an ABI 3130xl genetic analyzer with KB™ Basecaller software (Genomic Sequencing Center, Kingston, RI) then recombined into the pDEST17 destination vector to incorporate an N-terminal 6x-HIS epitope tag (Thermo Scientific) using DH5α as *E. coli* host. The final expression plasmid was verified by restriction endonuclease digestion and transformed into BL21 (CodonPlus) RIL competent cells (Agilent Technologies).

## Protein expression and purification

*E.coli* strains were grown in LB Broth supplemented with 100ug/ml ampicillin for positive selection. Bacterial overnight cultures (10 ml) were inoculated into 1L fresh broth and cultured at 37C for 2–2.5 hours to log phase ($OD_{600}$ of 0.5–0.7). Protein expression was initiated with addition of 0.5 mM IPTG at which point the temperature was shifted to 20˚C. Cultures were incubated for ~18–20 hours, centrifuged, and cell pellets were stored at -80˚C until purification. Recombinant NAMPT protein was lysed via sonication in buffer containing 50mM Tris pH8.0, 300mM NaCl, 10mM imidazole, and 5% glycerol then supplemented with 2mM DTT and protease inhibitors (Roche). Protein lysate was applied to 5ml of Ni-NTA resin (Quiagen) in a gravity drip format, washed with lysis buffer to remove weakly bound protein, and eluted in lysis buffer supplemented with 200mM and 350mM imidazole. SDS-PAGE analysis confirmed protein elution, that were then concentrated and injected onto a S200 16.60 size exclusion column equilibrated with buffer containing 50mM Tris pH8.0, 300mM NaCl, 5% glycerol and 2mM DTT and using AKTA Pure instrumentation (Cytiva Life Sciences). Final protein purity was >95% as defined by SDS-PAGE with Coomassie Staining and protein concentration was determined by $A_{280}$ absorption on a NanoDrop 2000c spectrophotometer (Thermo Scientific), using a MW of 58.6 kD and extinction coefficient of 83,200 $M^{-1}$ $cm^{-1}$. Recombinant NAMPT was stored at -80C in 50mM Tris, 300mM NaCl, 2mM DTT and >25% glycerol.

## NAMPT activity assay

The NAMPT activity assay was a modification of the protocol described by Zhang et. al [14]. Briefly, purified NAMPT was mixed with the given concentrations of Nicotinamide (NAM) and reaction buffer containing >0.4 mM phosphoribosylpyrophosphate (PRPP), 2mM ATP, 0.02% Bovine Serum Albumin, 2 mM dithiothreitol (DTT), 12 mM $MgCl_2$ and 50 HEPES Buffer (pH 8.0) to a final volume of 25 ul. The mixture was incubated at 37˚C for 15 minutes. The nicotinamide mononucleotide (NMN) product was detected following conversion to a fluorescent derivative as follows: addition of 10ul 2M KOH and 10 ul of 20% acetophenone in DMSO, incubation for 2 minutes on ice, addition of 45 ul 88% formic acid and incubation at 37˚C for 10 minutes. The reaction was then transferred to black flat-bottom 96-well plates (Grenier) for fluorescence detection on a Synergy HTX microplate reader (BioTek) at 360/40 nm excitation and 460/40 nm emission wavelengths (Fig 1).

## NAMPT modulator assay

The $EC_{50}$ values for potential NAMPT modulators were determined following the protocol for NAMPT assay with subsequent modifications. Potential modulators were incubated (0.5 ul of

appropriate stock concentration in DMSO to achieve the described final concentrations or 0.5 ul of freshly generated fruit juice/infusion) with 20 ul of reaction buffer for 30 minutes at 37°C. Enzyme reactions were initiated by addition of 4.5 ul of NAM (final concentration of 5uM). Molecules resulting in greater than 50% modulation of NAMPT activity were assessed to determine corresponding $EC_{50}$ values by evaluating NAMPT activity.

### Data analysis

All experiments were completed in triplicate and repeated as three independent trials, unless indicated otherwise. Each final value is reported as the mean ± standard deviation (SD). $EC_{50}$ values were determined by Prism 8.0 software from GraphPad via analysis using a non-linear regression with five parameters variable slope. Percent activity data was calculated with equation: Activity % = $(F_{RXN}−F_{RXN\ 0})$ / $(F100\%—F_0)$ where $F_{RXN}$ contains the modulating substance and NAM substrate, $F_{RXN\ 0}$ contains the modulating substance and water, $F_{100\%}$ contains DMSO and NAM, and $F_0$ contains DMSO and water.

## Results and discussion

### Computational ranking of NCI Diversity Set III and screening of 90 unique carbon skeletons generated 2 potential activators of NAMPT

Potential lead compounds were identified through a computational screen of the National Cancer Institute, Open Chemical Repository Diversity Set III using the freeware Autodock Vina. NCI Developmental Therapeutics Program (DPT) maintains a diverse set of 1597 carbon skeletons that were generated from almost 140,000 compounds amenable to forming structure-based hypotheses and are available for distribution. To generate the Diversity Set III, a large library of potential pharmacophores was screened to identify molecules that are relatively rigid, with 5 or fewer rotatable bonds, have a tendency to be planar, have a maximum of 1 chiral center, and contain pharmacologically desirable features, such as an absence of obvious leaving groups, no weakly bonded heteroatoms, and no organometallics [22]. The library contains both synthetically generated compounds and naturally derived compounds that display a diverse array of functional groups with the potential to interact in a biological setting.

The Diversity Set III chemical structures were downloaded from the NCI website (https://wiki.nci.nih.gov/display/NCIDTPdata/Chemical+Data) in SMILES format (.smi) then converted to 3D (.sdf) versions prior to docking into the NAMPT dimer (2GVG) search space centered around Lysine residue, K389, shown in Fig 2A. Docking was completed through the PyRx freeware, a Graphic User Interface for Autodock Vina. FK866, a potent inhibitor of NAMPT, was incorporated into the Diversity Set III library as a binding control within the active site. However, at the time of the enzymatic docking procedure, there were no known activators of NAMPT and thus no activating molecules were included in the virtual library.

Prior to docking the NCI library, the docking protocol was validated by removing incorporated ligands (known inhibitor, FK866 and enzymatic product, NMN) from published crystallographic structures PDBID: 2GVG and 2GVJ [10]. Each ligand was independently redocked into as active site channel of 2GVG to confirm the accuracy of the docking protocol. Overlays of the docked versions compared to the crystallographic structures are shown in Fig 2B. RMSD comparisons of the crystallographic and redocked structures were completed (Fig 2C), confirming the authenticity of the experimental design.

K389 was chosen as the search space for two reasons. First, the K389 residue of each monomer aligns in the dimer, spanning the dimerization plane between the openings of the active site channels. This dimer surface is concave, suggesting a possible location for protein-protein

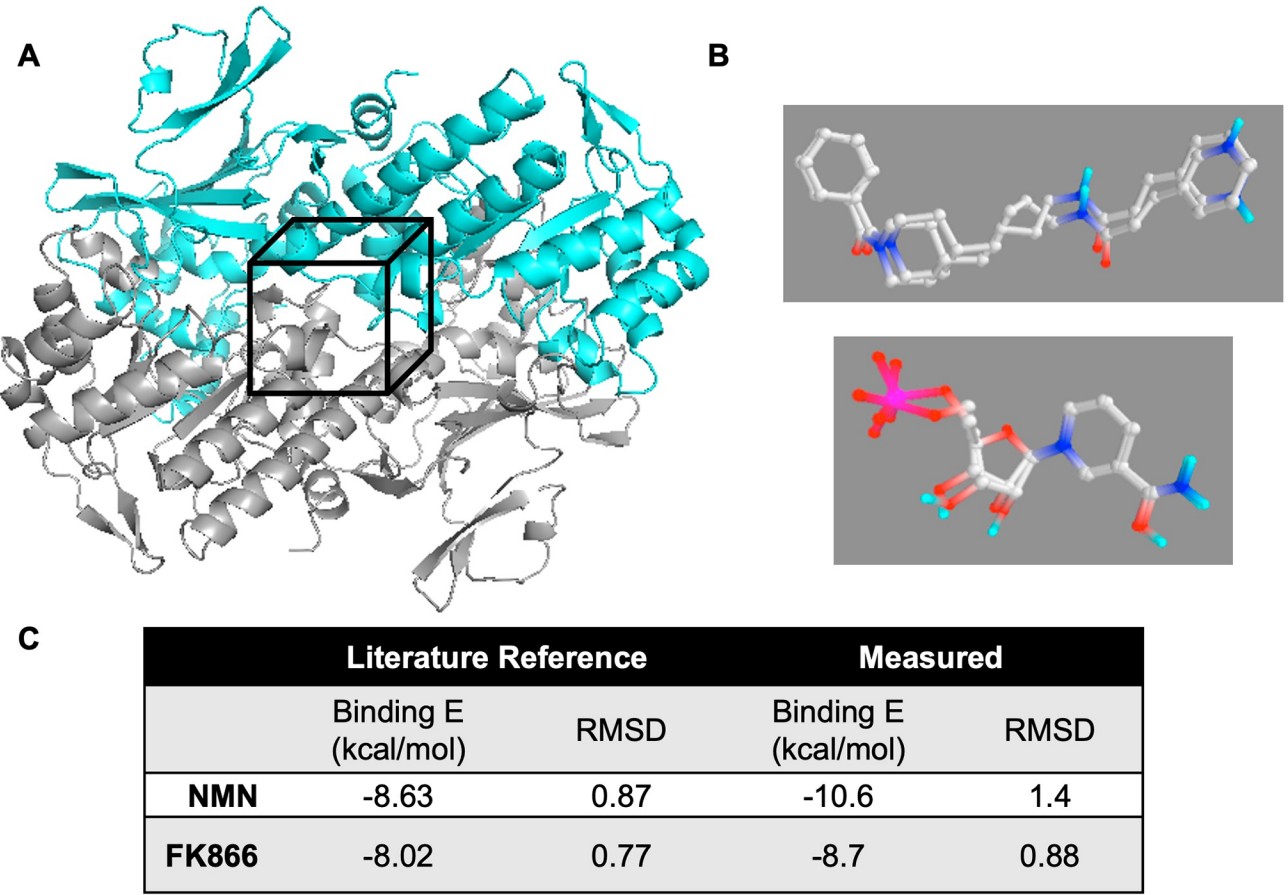

**Fig 2. Validation of Autodock Vina screening of NCI diversity set III.** (A) Ribbon image of NAMPT dimer (PDB ID: 2GVG) Black box indicates search space centered around K389 residue. (B) Overlays of Autodock redocking compared to interactions reported in 2GVG molecule. Top panel shows known inhibitor FK866. NAMPT product NMN shown in bottom panel. (C) Table comparing Binding Energy and Root Mean Square Deviation (RMSD) reported in literature to Autodock measured parameters.

|        | Literature Reference | | Measured | |
| --- | --- | --- | --- | --- |
|        | Binding E (kcal/mol) | RMSD | Binding E (kcal/mol) | RMSD |
| **NMN** | -8.63 | 0.87 | -10.6 | 1.4 |
| **FK866** | -8.02 | 0.77 | -8.7 | 0.88 |

interactions or allosteric binding of enzyme modulators [1]. Second, K389 has been reported as a possible ubiquitinylation Post-Translational Modification site [23], thus binding at this site may alter post-translational regulatory function.

Each member of the docked library was ranked and sorted according to the strongest energy binding pose presented. Those with the highest affinity for the target search space were compared with a similar ranking created for the random search space. Binding energies of the known inhibitor, FK866, within the active site was used as a cutoff binding affinity, generating a large pool of potential modulators to screen. All molecules binding at the determined cutoff and appearing on both lists were eliminated as potential lead compounds. Samples of lead compounds were obtained from the NCI/DPT Open Chemical Repository (http://dtp.cancer.gov).

Each potential lead compound was evaluated in a biochemical assay modified from Zhang et al [14]. Briefly, purified recombinant NAMPT was incubated at 37°C with 20uM of each compound in reaction buffer containing substrate PRPP. The enzymatic reaction was initiated with the addition of 5uM NAM. Enzymatic product, NMN, was converted into a fluorescent derivative through an in situ organic chemical reaction by the addition of KOH, acetophenone and formic acid, Fig 1. Each data point was completed in triplicates. Reaction plates were excited at 360 nm and fluorescence monitored at 460 nm. All control experiments were

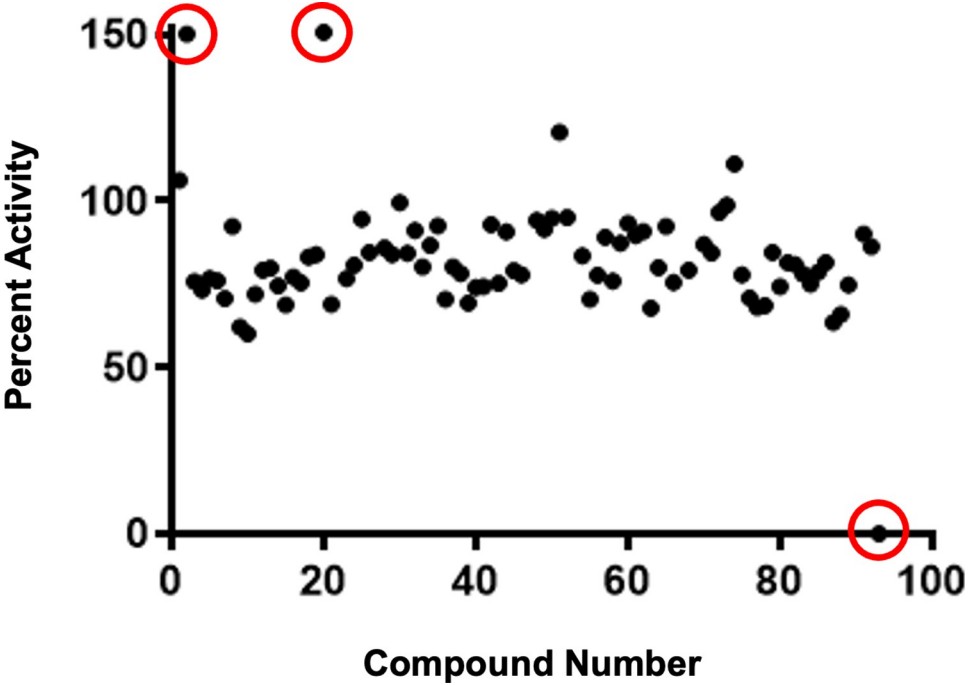

**Fig 3. NAMPT activity assay biochemical screen of Autodock Vina ranked carbon skeletons.** Unique, high binding molecules within the NCI Diversity Set III library screen were tested for modulation of NAMPT activity. Candidates were diluted to 20uM final concentration and evaluated a minimum of 2 independent trials. Molecules that consistently produced greater NMN product are circled (left of panel). Known inhibitor FK866 was included as a negative control, circle to bottom right.

completed and were reproduced similar to those reported [14]. To ensure that the recorded fluorescence was a function of NMN product formation and not intrinsic fluorescence from our lead compounds, the fluorescence of each lead compound was incorporated into the percent activity calculation. Each reported percent activity includes the baseline fluorescence of the buffer without either the candidate compound or the substrate NAM ($F_0$), the fluorescence of the candidate compound without substrate NAM ($F_{RXN\ 0}$), and the fluorescence of the NMN product without exposure to the candidate compound ($F_{100}$). Fig 3 compiles the assay results of 2 independent trials of each of 90 screened compounds, To normalize for each candidate compound's intrinsic fluorescence, we report a percent activity that removes background fluorescence prior to dividing by NAMPT activity in the absence of the candidate compound, see methods. The FK866 was used as a control and is highlighted in the red circle, bottom right of Fig 3. The majority of lead compounds resulted in activity between 60–120% of the NAMPT activity without additional compounds. Two candidates, however, routinely and reproducibly produced an increased fluorescence, red circles top left Fig 3, implying stimulation of enzymatic activity. Interestingly, both of these candidates are polyphenolic in nature although they exhibit a dramatically different carbon skeleton. Compound 2 corresponded to NSC19803 and compound 20 to NSC9037 within the diversity set library. Both were evaluated further to confirm the effect.

## Polyphenolic compound 20 (NSC9037) activates human NAMPT enzyme

Xanthen-3-one ring structures are known for their fluorescent qualities and are commonly used as diagnostic tools in the biosciences and earth sciences [24], the most widely known

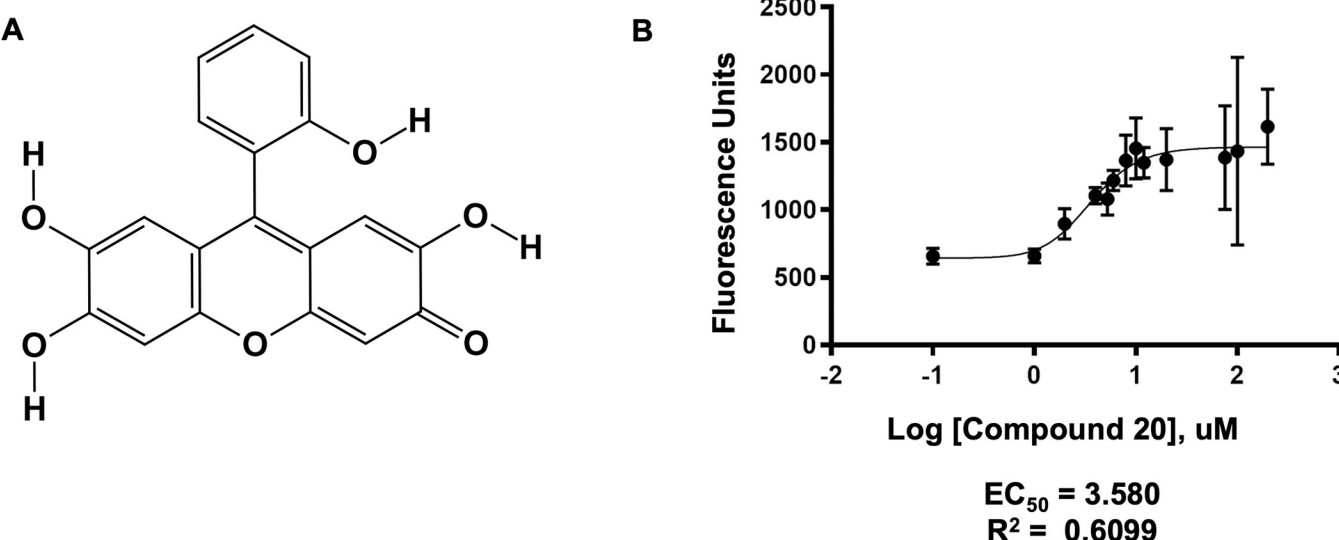

**Fig 4. Novel NAMPT activator compound 20.** (A) Organic structure of polyphenolic compound 20, 9-hydroxyphenylfluoron (NSC ID 9037, PubChem CID: 72722). (B) NAMPT activity as a function of increasing concentrations of compound 20. Graph represents the average of three independent trials each containing three replicates per data point. Data analyzed using the GraphPad Prism 8.0 software with a non-linear regression, asymmetric sigmoidal, 5PL equation and fitted with the least-squares method.

example is fluorescein [25] and its derivatives [26]. Hydroxylated derivatives have been reported to interact with HIV-1 nucleocapsid-p7 protein [27], as biological antioxidants [28] and as potential modulators of proliferation in tumor cell lines [29]. 9-hydroxyphenyl-fluoron, also known as NSC9037 (Fig 4A), is a polyphenolic xanthen-3-one compound that has recently been shown to inhibit the ubiquitination of Proliferating Cell Nuclear Antigen (PCNA), which initiates DNA damage tolerance pathways [30]. NSC9037 is also reported to exhibit antibacterial properties through its interaction with single-stranded binding protein in eubacteria [31].

In this study, the enzymatic activity of NAMPT increases as a function of increased exposure to NSC9037 (compound 20), generating as much as twice the amount of NMN product and exhibiting an $EC_{50}$ value of 3.580 uM, Fig 4B. It should be noted that compound 20 has an intrinsic fluorescence that overlaps with the fluorescent derivative used to monitor NAMPT activity. This background fluorescence can be seen in the increased standard deviation error at higher compound 20 concentrations. Nevertheless, the signal to noise ratio was sufficient to accurately evaluate the fluorescence generated solely from the NMN derivative.

To further evaluate the effect of compound 20 on the activity of the NAMPT enzyme, we titrated the substrate NAM in the presence and absence of 20uM compound 20, Fig 5, which clearly shows an increased maximum activity, 15,000 fluorescence units when compound 20 is present compared to 8,000 when absent. This corresponds to an increased $EC_{50}$ value, 7.526 uM with compound 20 compared to 6.052 uM in its absence.

## Polyphenolic compound 2 (NSC19803) activates human NAMPT enzyme

NSC19803, compound 2, is another polyphenolic compound. NSC19803 is the natural product myricitrin. Myricitrin is within the flavonoid family class of polyphenols and can be isolated from the root bark of the bayberry shrub native to Northern and Central America (Myrica cerifera). Myricitrin is also found in a wide range of plants worldwide, ranging from the Egyptian Lotus (Myrica esculenta) to the common Knotweed (Polygonum Aviculare) [32].

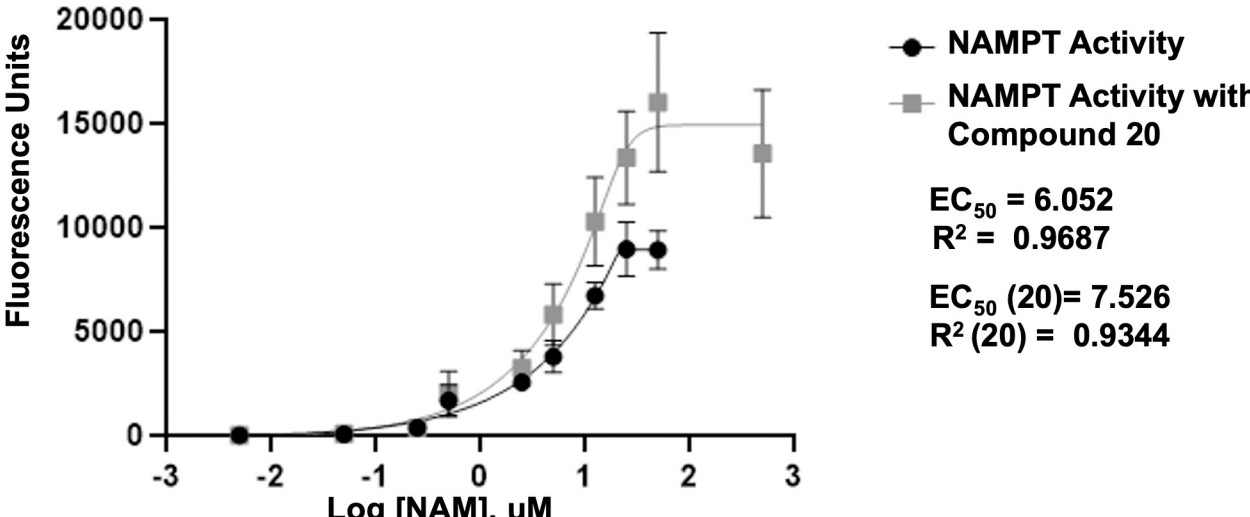

**Fig 5. NAMPT activity as a function of substrate NAM concentration.** Black circles represent NAMPT activity without compound 20 present while gray squares are with 20uM of compound 20 present. Each graph represents the average of three independent trials each containing three replicates per data point. Data was analyzed using the GraphPad prism software with a non-linear regression, asymmetric sigmoidal, 5PL equation and fitted with the least-squares method.

The organic structure of compound 2 is shown in Fig 6A, while its effect on the activity of the NAMPT enzyme is represented in Fig 6B. The maximum NMN product generated by NAMPT in the presence of uM concentrations of compound 2 produced greater that 1500

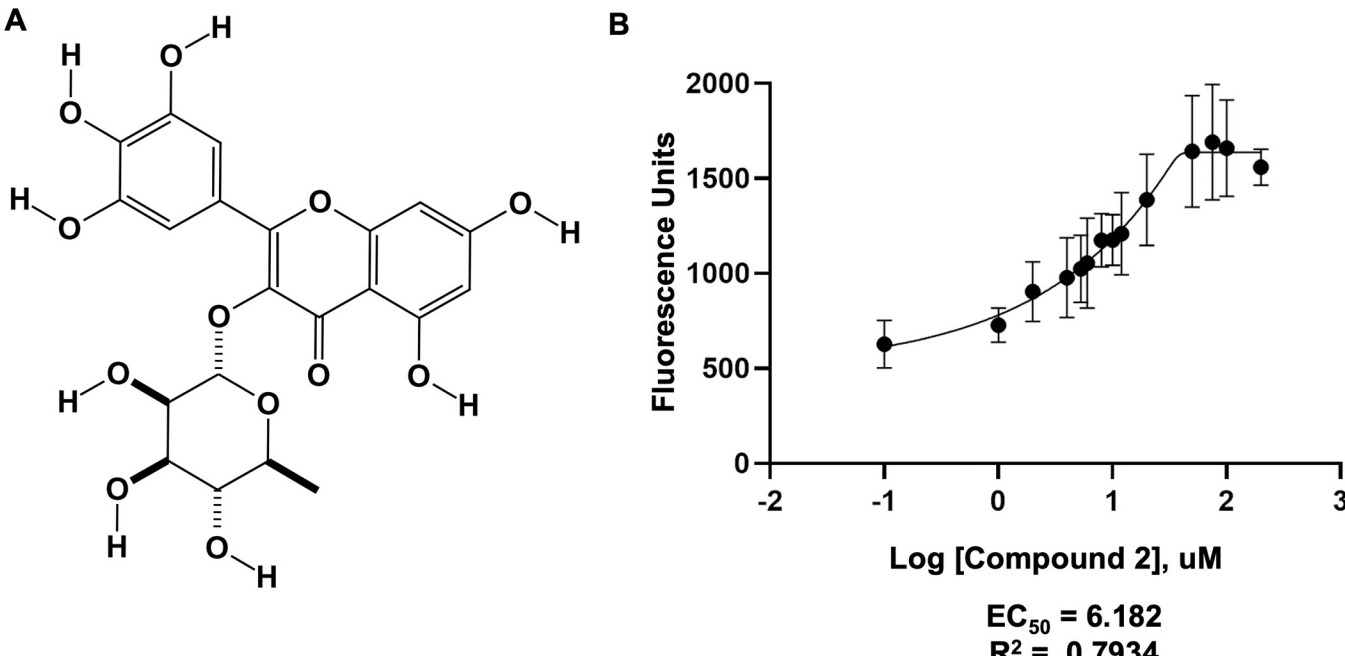

**Fig 6. Novel NAMPT activator compound 2.** (A) Organic structure of polyphenolic compound 2, Myricitrin (NSC ID 19803, PubChem CID 5281673). (B) NAMPT activity as a function of increasing concentrations of compound 2. Graph represents the average of three independent trials each containing three replicates per data point. Data was analyzed using the GraphPad prism software with a non-linear regression asymmetric sigmoidal, 5PL equation and fitted with the least-squares method.

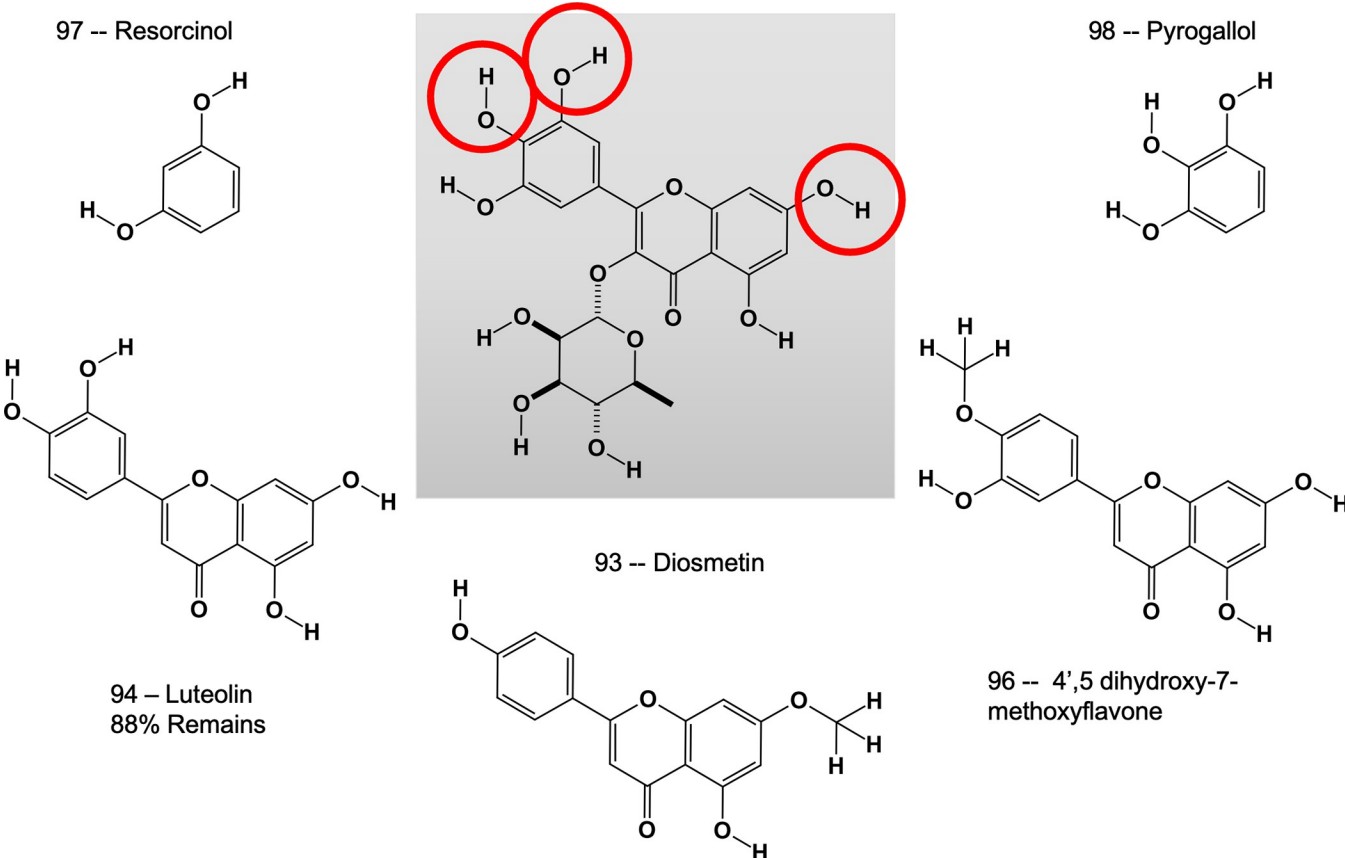

**Fig 7. Derivatives of compound 2 highlight important functional groups.** Compound 2 is shown in the center panel with critical hydroxyl groups circled. Derivatives that were tested for their ability to modulate NAMPT activity are shown surrounding compound 2.

fluorescence units while the baseline generation of NMN without compound 2 produced only 600 units. This suggests a significant increase in NMN product generated as a function of the addition of compound 2, with an $EC_{50}$ of 6.182 uM.

Multiple derivatives and fragments of compound 2 were tested to identify functional groups that are important in the interaction, Fig 7. Removal of the carbohydrate moiety from carbon 3 and the hydroxyl group on carbon 5' of the phenolic ring did not significant decrease the activation effect as shown by the increased NAMPT product generated upon addition of luteolin (Figs 7 and 8, compound 94). The conversion of hydroxyl group to a methoxyl group on carbon 7 and removal of the hydroxyl group on carbon 3' as shown in Diosmetin reversed the effect (Figs 7 and 8, compound 93). Similar results were seen when the hydroxyl group on carbon 4' was converted to a methoxy group (Figs 7 and 8, compound 96) Furthermore, simple polyphenol ring structures such as resorcinol and pyrogallol are not sufficient to produce the effect (Figs 7 and 8, compound 97 and 98). Taken together, this suggests that the hydroxyl groups associated with carbon 7 of the fused ring structure and carbons 3' and 4' of the phenolic ring (Fig 7, circles) are important for the activation of NAMPT.

## Phytochemicals found in food products activate NAMPT

Myricitrin is also found in the fruit products of the highbush blueberry (*Vaccinium corymbusum*) along with similar structural flavonoids such as Quercetin and Kaempferol [33].

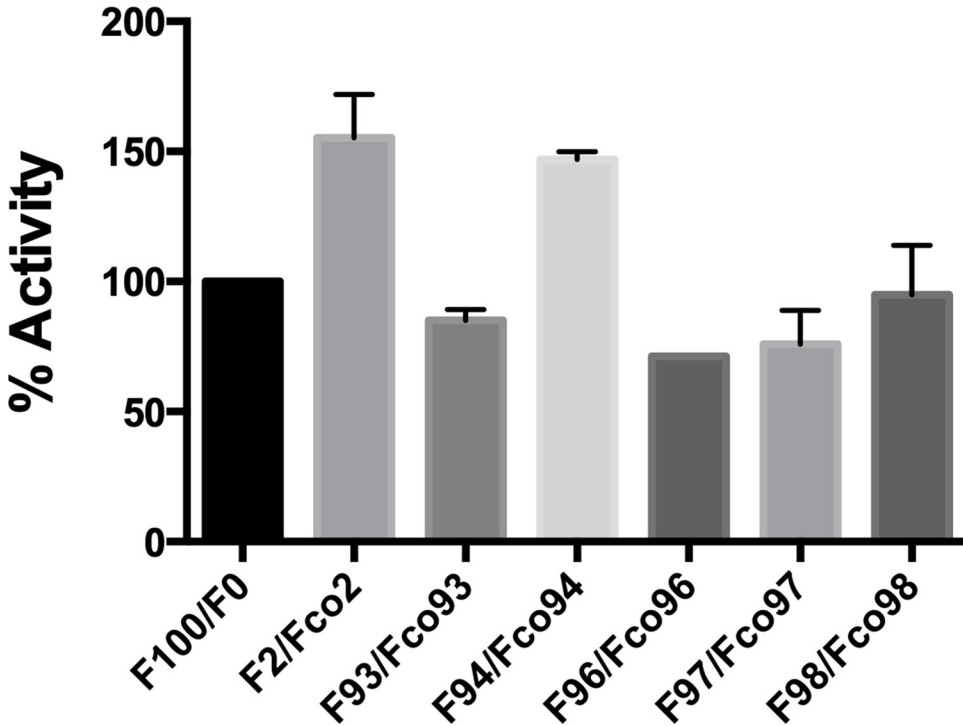

**Fig 8. Luteolin derivative of compound 2 activates NAMPT.** NAMPT activity upon exposure to 20uM of indicated compound. Compound 2 is shown as positive control, no modulating compound addition (F100/F0) acts as negative control. Graph represents the average of three independent trials each containing three replicates per data point.

Furthermore, commonly consumed fruits and seeds also contain high concentrations of the flavonoid subclasses, flavonols and anthrocyanins, examples include cranberry (*Vaccinium macrocarpon*) and cocoa (*Theobroma cacao*).

To investigate the hypothesis that phytochemicals found in natural products modulate the activity of NAMPT, we tested a variety of commonly consumed foods. Because plants have analogous biochemical systems to generate NAD, we first confirmed that 0.5uL of juice/infusion without recombinant NAMPT was not sufficient to convert NAM to the NMN product, Fig 9A. However, recombinant NAMPT exposed to identical quantities of some fruit juices or infusions produce more NMN than those lacking the exposure. Fig 9B clearly shows that several substances, including the juice from raw blueberries, cranberries, and cherries, as well as infusions of green tea and cocoa, positively influence the activity of NAMPT while strawberry juice and red wine did not significantly alter NAMPT activity.

## Autodock Vina suggests possible new interaction site of small-molecule activators

Three search spaces were tested prior to initiating the biochemical screen. Each space included a 25 cubic Angstrom box centered around either K229, K389, S199, or a random surface for comparison. As shown in Table 1, the random search space generated weaker interactions for both compound 20 and 2. Although compound 20 interacts strongly with each of the chosen search spaces, K389 was pursued because S199 is embedded into the interior of the active NAMPT molecule, and thus is not available within the active enzyme for molecular interactions [10]. Additionally, S199 in combination with S200 is essential for the dimerization and thus enzymatic function [34]. K229 is on the surface of the active NAMPT enzyme and is

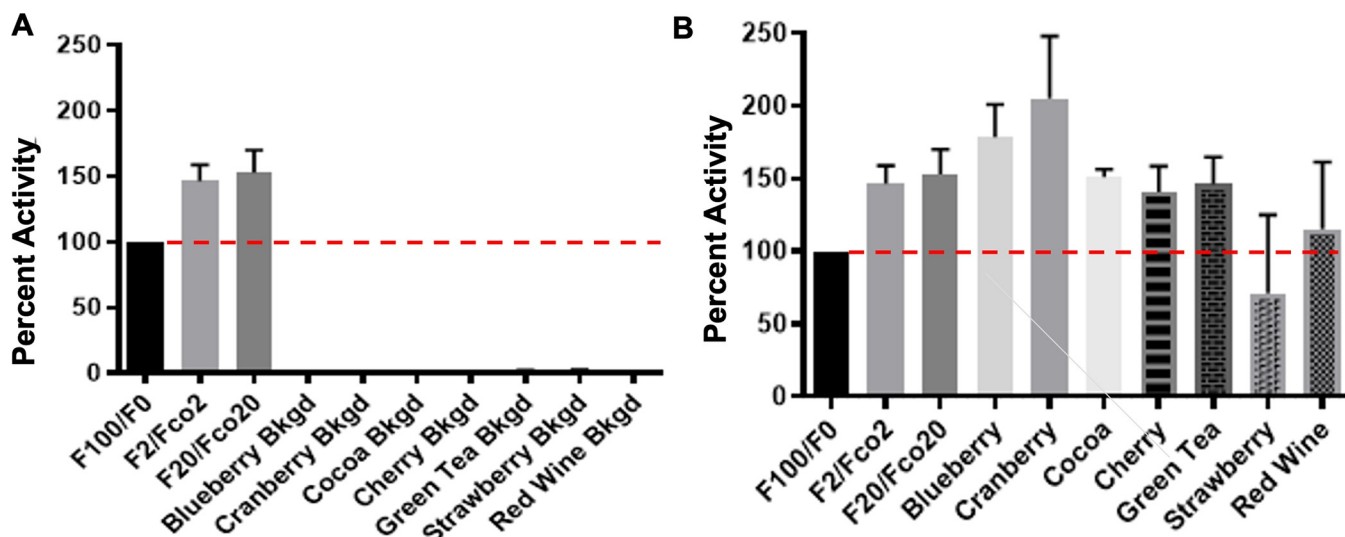

**Fig 9. NAMPT activation by natural products.** (A) NAMPT Activity background control, using only natural products as the source of NAMPT enzyme. (B) NAMPT activity using recombinant human NAMPT enzyme supplemented with natural products. Each bar represents the average of three independent trials, normalized for intrinsic fluorescence of natural product.

reported as a possible post-translational modification (PTM) site. K229 residues on each monomer align to span a surface cavity away from the active site channels and therefore interaction here could be possible for compound 2. Meanwhile both activators bound well to the K389 location and stimulated NAMPT to a similar degree.

Compound 2 binds best to the K389 search space. At this location, the strand containing K389 of one monomer aligns to its complement on the corresponding monomer when creating the active dimeric NAMPT. This creates a potential binding surface that lies within a concavity that encompasses portions of both active site channels and spans the dimerization plane. Concavities such as this are known to facilitate protein-protein interactions as well as containing ligand binding sites for small-molecule interactions [35]. K389 residues of each monomer lie at the center of the cavity and sit between the openings of both active site channels. The search space centered at K389 encompassed most, if not all, of the NMN product binding site within 2GVG, yet Autodock computations placed Compound 2 and Compound 20 in a similar location sandwiched between the K389 residues, in other words outside the active site channel but within the concavity. K229 appears to be a stronger binding site than K389 for compound 20, suggesting the possibility of unique binding locations for the two diverse carbon structures. Thus, it is possible that compound 20 and compound 2 interact in unique locations.

Fig 10 is a molecular visualization of possible binding interactions between NAMPT at the K389 cavity and compounds 20 and 2 [36]. Hydroxyl groups on compound 20 are within hydrogen bonding distance to potential binding partners K389, L390, and T391 on monomer

**Table 1. Autodock calculations for binding to NAMPT search spaces.**

|  | Search Space Binding Affinity (kcal/mol) | | | | |
| --- | --- | --- | --- | --- | --- |
|  | Random | K389 | K229 | S199 | △ K389 to Random |
| Cmpd 20 (9037) | -6.8 | -8.4 | -9.2 | -8.3 | -1.6 |
| Cmpd 2 (19803) | -7.4 | -8.9 | -7.7 | -7.7 | -1.5 |

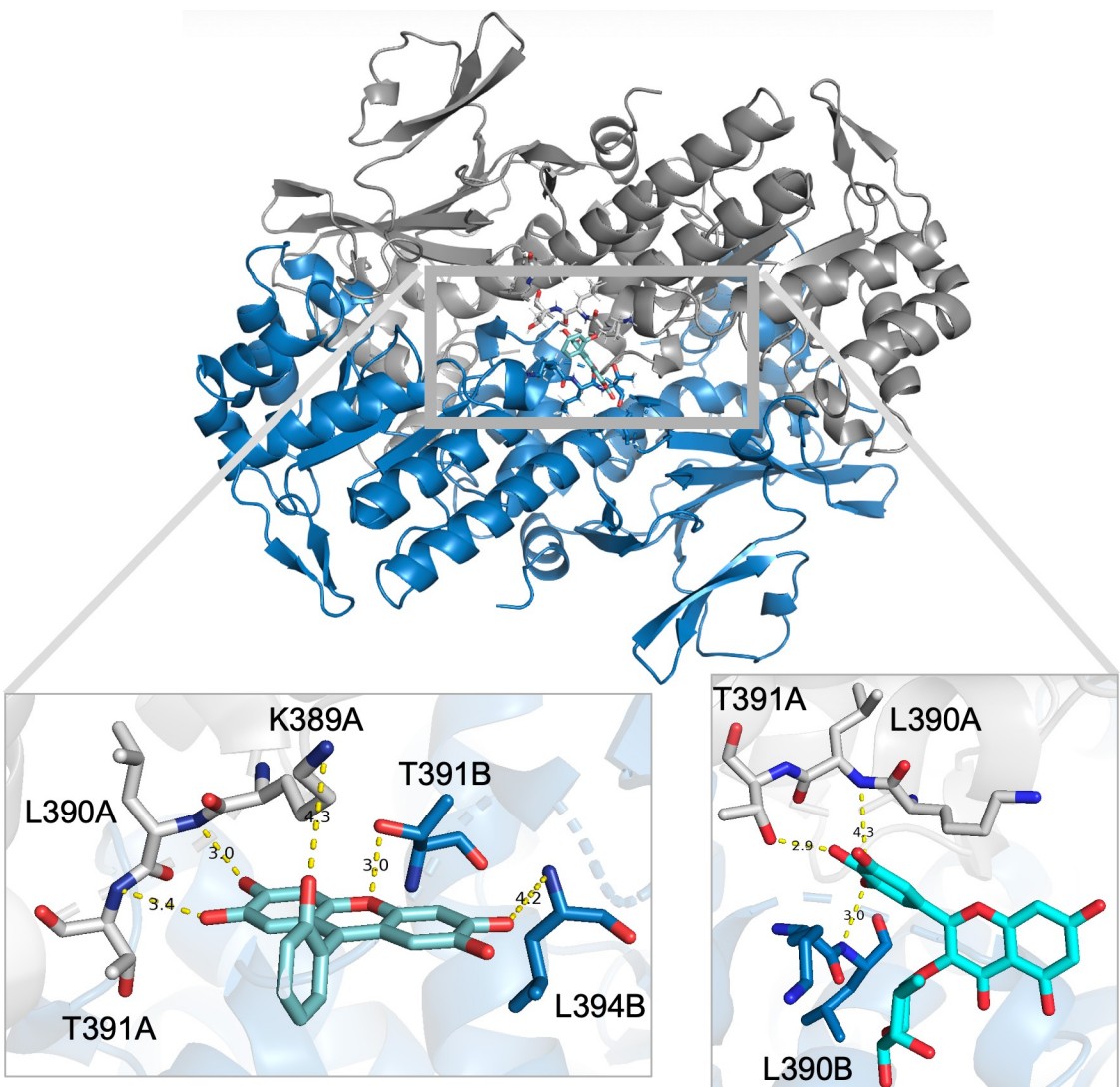

**Fig 10. Theoretical interaction between NAMPT K389 cavity and novel activators.** Molecular images were visualized using the educational version of PyMOL version 1.2 from Schrodinger, LLC. in the context of an undergraduate research experience. The active NAMPT dimer is represented in the top panel, with the K389 cavity highlighted in the gray box. Autodock Vina results of compound 20 docked into the search space suggests hydrogen bonding between compound 20 and individual NAMPT monomer units. (Left Panel) Autodock Vina results of compound 2 docked into the search space suggests hydrogen bonding between compound 2 and individual NAMPT monomer units. (Right Panel) There is no suggestion that the molecules will bind simultaneously.

A as well as T391 and L394 from monomer B, Fig 10 (left panel). Potential binding of compound 2 shows possible hydrogen bonding with L390 and T391 of monomer A but interactions on monomer B use L390, Fig 10, right panel. To be clear, it is unlikely that compound 2 and compound 20 bind to NAMPT at the same time, although this possibility was not specifically tested. Both compound 2 and compound 20 increase the product formation of NAMPT and appear to be binding along the same surface. Thus, compound 2 and compound 20 may affect NAMPT activity through a similar mechanism.

Binding at this location could influence NAMPT activity through numerous mechanisms, including stabilizing the active dimer formation, inducing allosteric conformational changes

to the region (ie, the opening to the active site channel to allow for easier binding of substrate and/or release of product), or as was suggested previously [17] but less likely since Autodock placed these molecules outside the active site, through shifting the equilibrium toward formation of NMN product and diminishing feedback inhibition. Further studies are required to investigate the exact mechanism by which these molecules increase the activity of NAMPT and thus increase cellular NAD.

## Conclusions

Given the importance of NAD within cellular systems and its necessity for life itself, understanding the regulation of NAD becomes crucial. NAMPT is the rate limiting step within the NAD salvage pathway, the predominant pathway for mammals, thus regulation of NAMPT affects global NAD levels.

There are several molecules reported to stimulate NAMPT activity. Many of these molecules are derivatives or share a similar molecular structure as the potent inhibitor FK866 whose mechanism of action is binding within the active site channel. Here, we report two novel carbon skeletons that increase product formation by recombinant NAMPT enzymes in biochemical assays. Unexpectedly both molecules shown to activate NAMPT production contain polyphenolic groups. Additionally, exposure to cranberry and blueberry juices produced a greater activity than either compound 2 or compound 20 alone, suggesting that other polyphenolic compounds found in these juices may also influence the activity of NAMPT.

NSC9037, compound 20, is a polyhydroxyl fluorescein derivative. In this study we report that micromolar quantities of Compound 20 increase NAMPT product production 2-fold. NSC19803, compound 2, is also a polyphenolic compound and is present in a wide range of natural products commonly consumed by humans, birds and animals. Exposure of NAMPT to micromolar quantities of Compound 2 also doubles the NMN product generated. Furthermore, we show significant increases in NAMPT product formation as a function of fruit juices and tea infusions, presumably due to a cocktail of compounds with similar structure to compound 2.

In silico analysis suggests that these compounds may interact with the K389 surface of the NAMPT dimer. This surface of NAMPT lies between the active site channels, spans the dimerization plane, and sits within a cavity that is reported to contain residues that are post-translationally modified. Clearly, more work is required to confirm the interaction domain of these compounds, including biophysical techniques to confirm the small-molecule interaction, x-ray crystallography to determine the exact binding location and/or site-directed mutagenesis to confirm individual residue binding partners.

## Supporting information

**S1 Data.**
(XLSX)

## Acknowledgments

The authors would like to thank Schrodinger, LLC for providing a freeware platform of PyMOL for undergraduate student to experience the power of molecular visualization.

## Author Contributions

**Conceptualization:** Karen H. Almeida.

**Data curation:** Karen H. Almeida, Kristen Chauvin.

**Formal analysis:** Karen H. Almeida, Lisbeth Avalos-Irving, Steven Berardinelli, Kristen Chauvin, Silvia Yanez.

**Funding acquisition:** Karen H. Almeida.

**Investigation:** Karen H. Almeida, Lisbeth Avalos-Irving, Steven Berardinelli, Kristen Chauvin, Silvia Yanez.

**Methodology:** Karen H. Almeida, Kristen Chauvin.

**Project administration:** Karen H. Almeida.

**Resources:** Karen H. Almeida.

**Supervision:** Karen H. Almeida.

**Validation:** Karen H. Almeida, Lisbeth Avalos-Irving, Steven Berardinelli, Kristen Chauvin, Silvia Yanez.

**Visualization:** Karen H. Almeida, Kristen Chauvin.

**Writing – original draft:** Karen H. Almeida.

**Writing – review & editing:** Karen H. Almeida, Lisbeth Avalos-Irving, Steven Berardinelli, Kristen Chauvin, Silvia Yanez.

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
