## [Decision Letter · Decision Letter 0]

20 Dec 2022

PONE-D-22-30462Novel carbon skeletons activate human NicotinAMide PhosphoribosylTransferase (NAMPT) enzyme in biochemical assayPLOS ONE

Dear Dr. Almeida,

Thank you for submitting your manuscript to PLOS ONE. After careful consideration, we feel that it has merit but does not fully meet PLOS ONE’s publication criteria as it currently stands. Therefore, we invite you to submit a revised version of the manuscript that addresses the points raised during the review process.

 Please submit your revised manuscript by Feb 03 2023 11:59PM. If you will need more time than this to complete your revisions, please reply to this message or contact the journal office at plosone@plos.org. Please include the following items when submitting your revised manuscript:A rebuttal letter that responds to each point raised by the academic editor and reviewer(s). You should upload this letter as a separate file labeled 'Response to Reviewers'.A marked-up copy of your manuscript that highlights changes made to the original version. You should upload this as a separate file labeled 'Revised Manuscript with Track Changes'.An unmarked version of your revised paper without tracked changes. You should upload this as a separate file labeled 'Manuscript'.

We look forward to receiving your revised manuscript.

Kind regards,

Qiu Sun

Academic Editor

PLOS ONE

Journal Requirements:

"Research reported in this publication was supported in part by grant subawards to KHA from the Rhode Island Institutional Development Award (IDeA) Network of Biomedical Research Excellence (https://web.uri.edu/riinbre/) from the National Institute of General Medical Sciences of the National Institutes of Health (https://www.nigms.nih.gov/) under grant number P20GM103430 and from the National Center for Research Resources of the National Institutes of Health under grant number 5P20RR016457-11.  Its contents are solely the responsibility of the authors and do not necessarily represent the official views of the NIGMS or the NIH.  Furthermore, this research is based in part upon work conducted using the Rhode Island Genomics and Sequencing Center that is supported in part by the National Science Foundation (https://www.nsf.gov/) under EPSCoR Grants Nos. 0554548 & EPS-1004057. The funders had no role in study design, data collection and analysis, decision to publish, or preparation of the manuscript."

"No competing interest exists."

Reviewers' comments:

Reviewer's Responses to Questions

**Comments to the Author**

1. Is the manuscript technically sound, and do the data support the conclusions?

Reviewer #1: Partly

Reviewer #2: Yes

2. Has the statistical analysis been performed appropriately and rigorously? 

Reviewer #1: N/A

Reviewer #2: Yes

3. Have the authors made all data underlying the findings in their manuscript fully available?

Reviewer #1: Yes

Reviewer #2: Yes

4. Is the manuscript presented in an intelligible fashion and written in standard English?

Reviewer #1: Yes

Reviewer #2: Yes

5. Review Comments to the Author

Reviewer #1: The manuscript titled “Novel carbon skeletons activate human NicotinAMide PhosphoribosylTransferase (NAMPT) enzyme in biochemical assay” by Almeida et al describes the results of a docking-based virtual screening coupled with biochemical assay aimed at finding novel compounds activating NAMPT, which is the rate-limiting enzyme in the NAD+ salvage pathway, regulates the NAD+ biosynthetic capacity of cells.

The concerns regarding the manuscript are uploaded as an attachment.

Reviewer #2: The paper presented the novel compounds which stimulates human NAMPT activities. The approach and methods are sound and writing is appropriate.

Minor point: His-tag protein (NAMPT) binding step was not described:Column size, kind of resin etc.

6. PLOS authors have the option to publish the peer review history of their article (what does this mean?). If published, this will include your full peer review and any attached files.

Reviewer #1: No

Reviewer #2: No

---

## [Author Response · Author response to Decision Letter 0]

31 Jan 2023

Response to Reviewers

PONE-D-22-30462_review

Novel carbon skeletons activate human NicotinAMide PhosphoribosylTransferase (NAMPT) enzyme in biochemical assay

Journal Requirements:

and 

I have completed this to the best of my ability.

"Research reported in this publication was supported in part by grant subawards to KHA from the Rhode Island Institutional Development Award (IDeA) Network of Biomedical Research Excellence (https://web.uri.edu/riinbre/) from the National Institute of General Medical Sciences of the National Institutes of Health (https://www.nigms.nih.gov/) under grant number P20GM103430 and from the National Center for Research Resources of the National Institutes of Health under grant number 5P20RR016457-11. Its contents are solely the responsibility of the authors and do not necessarily represent the official views of the NIGMS or the NIH. Furthermore, this research is based in part upon work conducted using the Rhode Island Genomics and Sequencing Center that is supported in part by the National Science Foundation (https://www.nsf.gov/) under EPSCoR Grants Nos. 0554548 & EPS-1004057. The funders had no role in study design, data collection and analysis, decision to publish, or preparation of the manuscript."

I have made the requested changes and included them in the Cover Letter.

"No competing interest exists."

I have included the amended Competing Interests Statement in the Cover Letter.

Review Comments to the Author

Reviewer #1: The manuscript titled “Novel carbon skeletons activate human NicotinAMide PhosphoribosylTransferase (NAMPT) enzyme in biochemical assay” by Almeida et al describes the results of a docking-based virtual screening coupled with biochemical assay aimed at finding novel compounds activating NAMPT, which is the rate-limiting enzyme in the NAD+ salvage pathway, regulates the NAD+ biosynthetic capacity of cells.

The concerns regarding the manuscript are as follows (in bold):

It is not clear which the NCI Diversity set was used for screening? III or IV? Is the screening library derived from only natural products? The authors should discuss that the both of virtual screening hits are polyphenolic compound?

The authors thank the reviewers for the opportunity to clarify our work. The NCI Diversity Set III was used as the docking library for the virtual screen. All reference to the Diversity Set IV has been removed from the manuscript.

The Diversity Set III screening library is a diverse set of potential pharmacophores and is not exculsively polyphenolic in nature. To help clarify, the authors have included the following on page 12 lines 273-275:

…The library contains both synthetically generated compounds and naturally derived compounds that display a diverse array of functional groups with the potential to interact in a biological setting. This library is not a collection of solely polyphenolic compounds.

Additionally, we have included a statement highlighting the result that out of >1500 compounds the two activators are both polyphenolic compounds. Page 14 lines 348-349:

…Interestingly, both of these candidates are polyphenolic in nature although they exhibit a dramatically different carbon skeleton.

Is there any information about the fluorescence intensity of the compounds (2 and 20) themselves? If not, should be checked.

The fluorescence of each individual modulator candidate was incorporated into the activity assay and described in the data analysis portion of the material and methods section, page 11 lines 256-258. It reads:

…Percent activity data was calculated with equation: Activity % = (FRXN – FRXN 0) / (F100% - F0 ) where FRXN contains the modulating substance and NAM substrate, FRXN 0 contains the modulating substance and water, F100% contains DMSO and NAM, and F0 contains DMSO and water.

Additionally, the following has been added to the results and discussion section, page 14, lines 335-341:

…To ensure that the recorded fluorescence was a function of NMN product formation and not intrinsic fluorescence from our lead compounds, the fluorescence of each lead compound was incorporated into the percent activity calculation. Each reported percent activity includes the baseline fluorescence of the buffer without either the candidate compound or the substrate NAM (F0), the fluorescence of the candidate compound without substrate NAM (FRXN 0), and the fluorescence of the NMN product without exposure to the candidate compound (F100). 

Although the studies on NAT revealed the critical role of K189 in boosting NAMPT activity, the binding modes of other activators with diverse scaffolds should also be considered and docking studies should be performed focusing on additional important residues (https://doi.org/10.1016/j.apsb.2022.07.016).

The authors thank the reviewers for directing us to an additional reference investigating the structural relationship of small molecule activation of NAMPT. We agree that the mechanism of action is still under investigation and performing additional docking studies focusing on the interaction between compound 2 and 20 at the proposed site of interaction will help to clarify the activation mechanism. This is the subject of a future investigation. 

We have included a reference to this study on page 6, lines 135-138 that includes

…However, structure-activity studies between NAMPT and a family of small molecule modulators suggests that increased product formation may be achieved through a critical water mediated interaction within the NAMPT active site but independent of K189. (20) Clearly the mechanism of NAMPT activation is still an open investigation.

Since the importance of water-mediated h-bonds in the interaction for both inhibitors and activators is known, water molecules in active site should be kept during docking simulations.

The Tang and Butterworth study (20) hypothesized that NAMPT activation by small molecules interaction is mediated by water molecule bridges within the active site. We agree that water should not be excluded however, this study was not available when our docking experiments were ongoing and thus any future work will include water molecules as potential interacting partners.

The NAMPT activators reported to date should be added to the screening library.

The authors agree that inclusion of the known activators of NAMPT would add to the study however, at the time of the screening, there were no known activators of NAMPT. Additionally, the undergraduate student completing the study has since graduated and is currently compleing a PhD program in a non-computational field. We have included the following explanation into the manuscript text. Page 12 lines 280-283

…FK866, a potent inhibitor of NAMPT, was incorporated into the Diversity Set III library as a binding control within the active site. However, at the time of the enzymatic docking procedure, there were no known activators of NAMPT and thus no activating molecules were included in the virtual library. 

The proposed binding modes of compounds 2 and 20 did not present the binding site properly. The figures need to be improved.

Figure 10 showing the proposed binding site has been improved to highlight the binding surface and represent the NAMPT:20 and NAMPT:2 interactions in a more standard format. The text and figure caption has been updated to reflect the new figüre, page 19-20, lines 532-554

The MD simulations should be performed for 2:NAMPT and 20:NAMPT complexes which would be support the findings.

The authors agree that MD simulations between 2:NAMPT and 20:NAMPT would support our findings. However, we do not feel that the absence of MD calculations precludes publication. We are currently collaborating with another undergraduate research lab to complete this analysis. 

Please check the words throughout the manuscrip, fig or Fig?

All reference to fig has been changed to Fig per PLOS One Formatting requirements.

Reviewer #2: The paper presented the novel compounds which stimulates human NAMPT activities. The approach and methods are sound and writing is appropriate.

Minor point: His-tag protein (NAMPT) binding step was not described:Column size, kind of resin etc.

We have included additional information on the His tagged purification of the NAMPT protein, page 9 lines 211-213:

…Protein lysate was applied to 5ml of Ni-NTA resin (Quiagen) in a gravity drip format, washed with lysis buffer to remove weakly bound protein, and eluted in lysis buffer supplemented with 200mM and 350mM imidazole.

---

## [Decision Letter · Decision Letter 1]

8 Mar 2023

Novel carbon skeletons activate human NicotinAMide PhosphoribosylTransferase (NAMPT) enzyme in biochemical assay

PONE-D-22-30462R1

Dear Dr. Almeida,

We’re pleased to inform you that your manuscript has been judged scientifically suitable for publication and will be formally accepted for publication once it meets all outstanding technical requirements.

Kind regards,

Qiu Sun

Academic Editor

PLOS ONE

Additional Editor Comments (optional):

Reviewers' comments:

Reviewer's Responses to Questions

**Comments to the Author**

1. If the authors have adequately addressed your comments raised in a previous round of review and you feel that this manuscript is now acceptable for publication, you may indicate that here to bypass the “Comments to the Author” section, enter your conflict of interest statement in the “Confidential to Editor” section, and submit your "Accept" recommendation.

Reviewer #1: All comments have been addressed

2. Is the manuscript technically sound, and do the data support the conclusions?

Reviewer #1: Yes

3. Has the statistical analysis been performed appropriately and rigorously? 

Reviewer #1: Yes

4. Have the authors made all data underlying the findings in their manuscript fully available?

Reviewer #1: Yes

5. Is the manuscript presented in an intelligible fashion and written in standard English?

Reviewer #1: Yes

6. Review Comments to the Author

Reviewer #1: The manuscript seems to be corrected and modified according to reviewer suggestions.

In my opinion, it might be now accepted for publication in PLOS ONE.

7. PLOS authors have the option to publish the peer review history of their article (what does this mean?). If published, this will include your full peer review and any attached files.

Reviewer #1: No

---

## [Editor Report · Acceptance letter]

20 Mar 2023

PONE-D-22-30462R1 

Novel carbon skeletons activate human NicotinAMide Phosphoribosyl Transferase (NAMPT) enzyme in biochemical assay 

Dear Dr. Almeida:

I'm pleased to inform you that your manuscript has been deemed suitable for publication in PLOS ONE. Congratulations! Your manuscript is now with our production department. 

Kind regards, 

on behalf of

Dr. Qiu Sun 

Academic Editor

PLOS ONE